# KNOWLEDGE HYPERGRAPHS: PREDICTION BEYOND BINARY RELATIONS

## ABSTRACT

Knowledge graphs store facts using relations between pairs of entities. In this work, we address the question of link prediction in knowledge hypergraphs where each relation is defined on *any number* of entities. While there exist techniques (such as reification) that convert the non-binary relations of a knowledge hypergraph into binary ones, current embedding-based methods for knowledge graph completion do not work well out of the box for knowledge graphs obtained through these techniques. Thus we introduce *HSimplE* and *HypE*, two embedding-based methods that work directly with knowledge hypergraphs in which the representation of an entity is a function of its position in the relation. We also develop public benchmarks and baselines for this task and show experimentally that the proposed models are more effective than the baselines. Our experiments show that *HypE* outperforms *HSimplE* when trained with fewer parameters and when tested on samples that contain at least one entity in a position never encountered during training.

## 1 INTRODUCTION

*Knowledge Hypergraphs* are graph structured knowledge bases that store facts about the world in the form of relations between two or more entities. They can be seen as one generalization of *Knowledge Graphs*, in which relations are defined on exactly two entities. Since accessing and storing all the facts in the world is difficult, knowledge bases are incomplete; the goal of *link prediction* (or *knowledge completion*) in knowledge (hyper)graphs is to predict unknown links or relationships between entities based on the existing ones. In this work we are interested in the problem of link prediction in knowledge hypergraphs. Our motivation for studying link prediction in these more sophisticated knowledge structures is based on the fact that most knowledge in the world has inherently complex composition, and that not all data can be represented as a relation between two entities without either losing a portion of the information or creating incorrect data points.

Link prediction in knowledge graphs is a problem that is studied extensively, and has applications in several tasks such as searching (Singhal, 2012) and automatic question answering (Ferrucci et al., 2010). In these studies, knowledge graphs are defined as directed graphs having nodes as entities and labeled edges as relations; edges are directed from the *head* entity to the *tail* entity. The common data structure for representing knowledge graphs is a set of triples $relation(head, tail)$ that represent information as a collection of binary relations. There exist a large number of knowledge graphs that are publicly available, such as NELL (Carlson et al., 2010) and FREEBASE (Bollacker et al., 2008). It is noteworthy to mention that FREEBASE is a complex knowledge base where 61% of the relations are beyond binary (defined on more than two nodes). However, current methods use a simplified version of FREEBASE where the non-binary relations are converted to binary ones (defined on exactly two entities).

Embedding-based models (Nguyen, 2017) have proved to be effective for knowledge graph completion. These approaches learn embeddings for entities and relations. To find out if $r(h, t)$ is a fact (i.e. is true), such models define a function that embeds relation $r$ and entities $h$ and $t$, and produces the probability that $r(h, t)$ is a fact. While successful, such embedding-based methods make the strong assumption that all relations are binary.

In this work, we introduce two embedding-based models that perform link prediction in knowledge hypergraphs. The first is *HSimplE*, which is inspired from *SimplE* (Kazemi & Poole, 2018), origi-

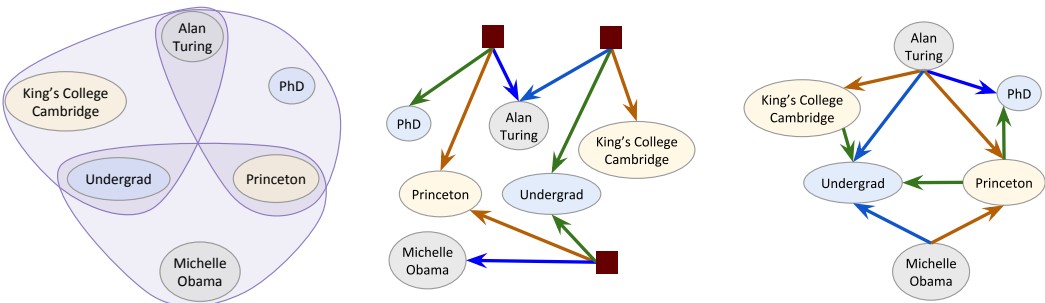

(a) `DEGREE_FROM_UNIVERSITY` defined on three facts.

(b) Reifying non-binary relations with three additional entities.

(c) Converting non-binary relations into cliques.

Figure 1: In this example, the three facts in the original graph (a) show that Turing received his PhD from Princeton and his undergraduate degree from King's College Cambridge. Figures (b) and (c) show two methods of converting this ternary relation into three binary ones.

nally designed to perform link prediction in knowledge graphs. For a given entity, *HSimplE* shifts the entity embedding by a value that depends on the position of the entity in the given relation. Our second model is *HypE*, which in addition to learning entity embeddings, learns positional (convolutional) filters; these filters are disentangled from entity representations and are used to transform the representation of an entity based on its position in a relation. We show that both HSimplE and HypE are fully expressive. To evaluate our models, we introduce two new datasets from subsets of FREEBASE, and develop baselines by extending existing models on knowledge graphs to work with hypergraphs. We evaluate the proposed methods on standard binary and non-binary datasets. While both HSimplE and HypE outperform our baselines and the state-of-the-art, HypE is more effective with fewer parameters. It also produces much better results when predicting relations that contain at least one entity in a position never encountered during training, demonstrating the clear advantage of disentangling position representation from entity embeddings.

The contributions of this paper are: (1) HypE and HSimplE, two embedding-based methods for knowledge hypergraph completion that outperform the baselines for knowledge hypergraphs, (2) a set of baselines for knowledge hypergraph completion, and (3) two new knowledge hypergraphs obtained from subsets of FREEBASE, which can serve as new evaluation benchmarks for knowledge hypergraph completion methods.

## 2 MOTIVATION AND RELATED WORK

Knowledge hypergraph completion is a relatively under-explored area. We motivate our work by outlining that adjusting current models to accommodate hypergraphs does not yield satisfactory results. Existing knowledge graph completion methods can be used in the beyond-binary setting by either (1) extending known models to work with non-binary relational data (e.g., m-TransH (Wen et al., 2016)), or by (2) converting non-binary relations into binary ones using methods such as reification or star-to-clique (Wen et al., 2016), and then applying known link prediction methods.

In the first case, the only example that extends a known model to work with non-binary relations is m-TransH (Wen et al., 2016), which is an extension of TransH (Wang et al., 2014), and which we show to be less effective than our models in Section 6. The second case is about restructuring a knowledge hypergraph to work with current knowledge graph completion methods. One common approach to reduce a hypergraph into a graph is *reification*. In order to reify a fact with a relation defined on $k$ entities, we first create a new entity $e$ (square nodes in Figure 1b) and connect $e$ to each of the $k$ entities that are part of the given fact. Another approch is *Star-to-clique*, which converts a fact defined on $k$ entities into $\binom{k}{2}$ facts with distinct relations between all pairwise entities in the fact. See Figure 1c.

Both conversion approaches have their caveats when current link-prediction models are applied to the resulting graphs. The example in Figure 1a shows three facts that pertain to the relation `DEGREE_FROM_UNIVERSITY`. When we reify the hypergraph in this example (Figure 1b), we

add three reified entities. At test time, we again need to reify the test samples, which means we need a way to embed newly created entities about which we have almost no information. Applying the star-to-clique method to the hypergraph does not yield better results either: in this case, the resulting graph loses some of the information that the original hypergraph had — in Figure 1c, it is no longer clear which degree was granted by which institution.

Existing methods that relate to our work in this paper can be grouped into three main categories:

**Knowledge graph completion.** Embedding-based models such as *translational* (Bordes et al., 2013; Wang et al., 2014), *bilinear* (Yang et al., 2015; Trouillon et al., 2016; Kazemi & Poole, 2018), and *deep models* (Nickel et al., 2011; Socher et al., 2013) have proved effective for knowledge graphs where all relations are binary. We extend some of the models in this category and compare them with the proposed methods.

**Knowledge hypergraph completion.** Soft-rule models (Richardson & Domingos, 2006; De Raedt et al., 2007; Kazemi et al., 2014) can easily handle variable arity relations and have the advantage of being interpretable. However, they have a limited learning capacity and can only learn a subset of patterns (Nickel et al., 2016). Embedding-based methods are more powerful than soft-rule approaches. Guan et al. (2019) proposed an embedding-based method based on the star-to-clique approach which its caveats are discussed. m-TransH (Wen et al., 2016) extends TransH (Wang et al., 2014) to knowledge hypergraph completion. Kazemi & Poole (2018) prove that TransH and consequently m-TransH are not fully expressive and have restrictions in modeling relations. In contrast, our embedding-based proposed models are fully expressive and outperform m-TransH.

**Learning on hypergraphs.** Hypergraph learning has been employed to model high-order correlations among data in many tasks, such as in video object segmentation (Huang et al., 2009) and in modeling image relationships and image ranking (Huang et al., 2010). There is also a line of work extending graph neural networks to hypergraph neural networks (Feng et al., 2019) and hypergraph convolution networks (Yadati et al., 2018). On the other hand, these models are designed for undirected hypergraphs, with edges that are not labeled (no relations), while knowledge hypergraphs are directed and labeled. As there is no clear or easy way of extending these models to our knowledge hypergraph setting, we do not consider them as baselines for our experiments.

## 3 DEFINITION AND NOTATION

A world consists of a finite set of entities $\mathcal{E}$, a finite set of relations $\mathcal{R}$, and a set of tuples $\tau$ defined over $\mathcal{E}$ and $\mathcal{R}$. Each tuple in $\tau$ is of the form $r(v_1, v_2, \ldots, v_k)$ where $r \in \mathcal{R}$ is a relation and each $v_i \in \mathcal{E}$ is an entity, for all $i = 1, 2, \ldots, k$. Here *arity* $|r|$ of a relation $r$ is the number of arguments that the relation takes and is fixed for each relation. A world specifies what is true: all the tuples in $\tau$ are true, and the tuples that are not in $\tau$ are false. A knowledge hypergraph consists of a subset of the tuples $\tau' \subseteq \tau$. Link prediction in knowledge hypergraphs is the problem of predicting the missing tuples in $\tau'$, that is, finding the tuples $\tau \setminus \tau'$.

An *embedding* is a function that converts an entity or a relation into a vector (or sometimes a higher order tensor) over a field (typically the real numbers). We use bold lower-case for vectors, that is, $\mathbf{e} \in \mathbb{R}^k$ is an embedding of entity $e$, and $\mathbf{r} \in \mathbb{R}^l$ is an embedding of a relation $r$.

Let $\mathbf{v_1}, \mathbf{v_2}, \ldots, \mathbf{v_k}$ be a set of vectors. The variadic function $\mathrm{concat}(\mathbf{v_1}, \ldots, \mathbf{v_k})$ outputs the concatenation of its input vectors. The 1D convolution operator $*$ takes as input a vector $\mathbf{v}$ and a convolution weight filter $\omega$, and outputs the convolution of $\mathbf{v}$ with the filters $\omega$. We define the variadic function $\odot()$ to be the sum of the element-wise product of its input vectors, namely $\odot(\mathbf{v_1}, \mathbf{v_2}, \ldots, \mathbf{v_k}) = \sum_{i=1}^{\ell} \mathbf{v_1}^{(i)} \mathbf{v_2}^{(i)} \ldots \mathbf{v_k}^{(i)}$ where each vector $\mathbf{v_i}$ has the same length, and $\mathbf{v_j}^{(i)}$ is the $i$-th element of vector $\mathbf{v_j}$.

For the task of knowledge graph completion, an embedding-based model defines a function $\phi$ that takes a tuple $x$ as input, and generates a prediction, *e.g.*, a probability (or score) of the tuple being true. A model is *fully expressive* if given any complete world (full assignment of truth values to all tuples), there exists an assignment of values to the embeddings of the entities and relations that accurately separates the tuples that are true in the world from those that are false.

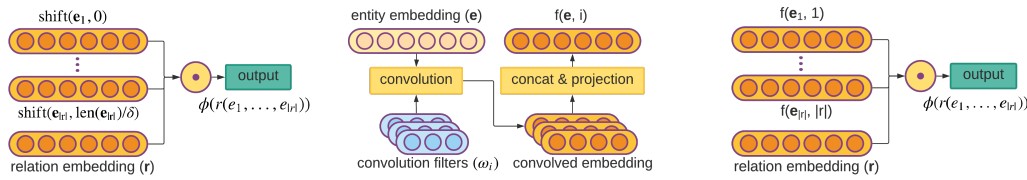

(a) Function $\phi$ for HSimplE.    (b) Function $f(\mathbf{e}, i)$ used in HypE.    (c) Function $\phi$ for HypE.

Figure 2: Visualization of HypE and HSimplE architectures. (a) function $\phi$ for HSimplE transforms entity embeddings by shifting them based on their position and combining them with the relation embedding. (b) function $f(\mathbf{e}, i)$ for HypE takes an entity embedding and the position the entity appears in the given tuple, and returns a vector. (c) function $\phi$ takes as input a tuple and outputs the score of HypE for the tuple.

## 4    KNOWLEDGE HYPERGRAPH EMBEDDING: PROPOSED METHODS

The idea at the core of our methods is that the way an entity representation is used to make predictions is affected by the role that the entity plays in a given relation. In the example in Figure 1, Turing plays the role of a student at a university, but he may have a different role (e.g. 'professor') in another relation. This means that the way we use Turing's embedding may need to be different for computing predictions for each of these roles.

The prediction for an entity should depend on the position it appears. If the prediction does not depend on the position, then the relation has to be symmetric. If it does and positions are learned independently, information about one position will not interact with that of others. It should be noted that in several embedding-based methods for knowledge graph completion, such as canonical polyadic (Hitchcock, 1927; Lacroix et al., 2018), ComplEx (Trouillon et al., 2016), and SimplE (Kazemi & Poole, 2018), the prediction depends on the position of an entity.

In what follows, we propose two embedding-based methods for link prediction in knowledge hypergraphs. The first model is inspired by SimplE and has its roots in link prediction in knowledge graphs; the second model takes a fresh look at knowledge completion as a multi-arity problem, without first setting it up within the frame of binary relation prediction.

**HSimplE:** HSimplE is an embedding-based method for link prediction in knowledge hypergraphs that is inspired by SimplE (Kazemi & Poole, 2018). SimplE learns two embedding vectors $\mathbf{e}^{(1)}$ and $\mathbf{e}^{(2)}$ for an entity $e$ (one for each possible position of the entity), and two embedding vectors $\mathbf{r}^{(1)}$ and $\mathbf{r}^{(2)}$ for a relation $r$ (with one relation embedding as the inverse of the other). It then computes the score of a triple as $\phi(r(e_1, e_2)) = \odot(\mathbf{r}^{(1)}, \mathbf{e_1}^{(1)}, \mathbf{e_2}^{(2)}) + \odot(\mathbf{r}^{(2)}, \mathbf{e_2}^{(1)}, \mathbf{e_1}^{(2)})$.

In HSimplE, we adopt the idea of having different representations for an entity based on its position in a relation, and updating all these representations from a single training tuple. We do this by representing each entity $e$ as a single vector $\mathbf{e}$ (instead of multiple vectors as in SimplE), and each relation $r$ as a single vector $\mathbf{r}$. Conceptually, each $\mathbf{e}$ can be seen as the concatenation of the different representations of $e$ based on every possible position. For example, in a knowledge hypergraph where the relation with maximum arity is $\delta$, an entity can appear in $\delta$ different positions; hence $\mathbf{e}$ will be the concatenation of $\delta$ vectors, one for each possible position. HSimplE scores a tuple using the following function:

$$\phi(r(e_i, e_j, \ldots, e_k)) = \odot(\mathbf{r}, \mathbf{e_i}, \mathrm{shift}(\mathbf{e_j}, \mathrm{len}(\mathbf{e_j})/\delta), \ldots, \mathrm{shift}(\mathbf{e_k}, \mathrm{len}(\mathbf{e_k}) * (\delta - 1)/\delta)).$$

Here, $\mathrm{shift}(\mathbf{a}, \mathrm{x})$ shifts vector $\mathbf{a}$ to the left by $x$ steps, $\mathrm{len}(\mathbf{e})$ returns length of vector $\mathbf{e}$, and $\delta = \max_{r \in \mathcal{R}}(|r|)$. We observe that for knowledge graphs ($\delta = 2$), SimplE is a special instance of HSimplE, with $\mathbf{e} = \mathrm{concat}(\mathbf{e}^{(1)}, \mathbf{e}^{(2)})$ and $\mathbf{r} = \mathrm{concat}(\mathbf{r}^{(1)}, \mathbf{r}^{(2)})$. The architecture of HSimplE is summarized in Figure 2a.

**HypE:** HypE learns a single representation for each entity, a single representation for each relation, and positional convolutional weight filters for each possible position. At inference time, the appropriate positional filters are first used to transform the embedding of each entity in the given fact; these transformed entity embeddings are then combined with the embedding of the relation to produce a *score*, *e.g.*, a probability value that the input tuple is true. The architecture of HypE is summarized in Figures 2b and 2c.

Let $n$, $l$, $d$, and $s$ denote the number of filters per position, the filter-length, the embedding dimension and the stride of the convolution, respectively. Let $\omega_i \in \mathbb{R}^n \times \mathbb{R}^l$ be the convolutional filters associated with an entity at position $i$, and let $\omega_{ij} \in \mathbb{R}^l$ be the $j$th row of $\omega_i$. We denote by $P \in \mathbb{R}^{nq} \times \mathbb{R}^d$ the projection matrix, where $q = \lfloor (d-l)/s \rfloor + 1$ is the feature map size. For a given tuple, define $f(\mathbf{e}, i) = \mathrm{concat}(\mathbf{e} * \omega_{i1}, \ldots, \mathbf{e} * \omega_{in})P$ to be a function that returns a vector of size $d$ based on the entity embedding $\mathbf{e}$ and it's position $i$ in the tuple. Thus, each entity embedding $\mathbf{e}$ appearing at position $i$ in a given tuple is convolved with the set of position-specific filters $\omega_i$ to give $n$ feature maps of size $q$. All $n$ feature maps corresponding to an entity are concatenated to a vector of size $nq$ and projected to the embedding space by multiplying it by $P$. The projected vectors of entities and the embedding of the relation are combined by an inner-product to define $\phi$:

$$\phi(r(e_1, \ldots, e_{|r|})) = \odot(\mathbf{r}, f(\mathbf{e_1}, 1), \ldots, f(\mathbf{e_{|\mathbf{r}|}}, |r|)) \tag{1}$$

The advantage of learning positional filters independent of entities is two-folds: On one hand, learning a single vector per entity keeps entity representations simple and disentangled from its position in a given fact. On the other hand, unlike HSimplE, HypE learns positional filters from all entities that appear in the given position; Overall, this separation of representations for entities, relations, and position facilitates the representation of knowledge bases having facts of arbitrary number of entities. It also gives HypE an advantage in the case when we test a trained HypE model on a tuple that contains an entity in a position never seen before at train time. We discuss this further in Section 6.1.

Both HSimplE and HypE are fully expressive — an important property that has been the focus of several studies (Fatemi et al., 2019; Trouillon et al., 2017; Xu et al., 2018). A model that is not fully expressive can easily underfit to the training data and embed assumptions that may not be reflected in reality. We defer the proofs of expressivity to Appendix A.

### 4.1 Objective Function and Training

To learn either of a HSimplE or HypE model, we use stochastic gradient descent with mini-batches. In each learning iteration, we iteratively take in a batch of positive tuples from the knowledge hypergraph. As we only have positive instances available, we need also to train our model on negative instances. For this purpose, for each positive instance, we produce a set of negative instances. For negative sample generation, we follow the contrastive approach of Bordes et al. (2013) for knowledge graphs and extend it to knowledge hypergraphs: for each tuple, we produce a set of negative samples of size $N|r|$ by replacing each of the entities with $N$ random entities in the tuple, one at a time. Here, $N$ is the ratio of negative samples in our training set, and is a hyperparameter.

Given a knowledge hypergraph defined on $\tau'$, we let $\tau'_{train}$, $\tau'_{test}$, and $\tau'_{valid}$ denote the train, test, and validation sets, respectively, so that $\tau' = \tau'_{train} \cup \tau'_{test} \cup \tau'_{valid}$. For any tuple $x$ in $\tau'$, we let $T_{neg}(x)$ be a function that generates a set of negative samples through the process described above. Let $\mathbf{r}$ represent relation embeddings, $\mathbf{e}$ represent entity embeddings, and let $\phi$ be the function given by equation 1 that maps a tuple to a score based on $\mathbf{r}$ and $\mathbf{e}$. We define the following cross entropy loss, which is a combination of softmax and negative log likelihood loss, and has been shown to be effective for link prediction (Kadlec et al., 2017):

$$\mathcal{L}(\mathbf{r}, \mathbf{e}) = \sum_{x' \in \tau'_{train}} -log\left( \frac{e^{\phi(x')}}{e^{\phi(x')} + \sum_{x \in T_{neg}(x')} e^{\phi(x)}} \right)$$

## 5 Experimental Setup

### 5.1 Datasets

We conduct experiments on a total of 5 different datasets. For the experiments on datasets with binary relations, we use two standard benchmarks for knowledge graph completion: WN18 (Bordes et al., 2014) and FB15k (Bordes et al., 2013). WN18 is a subset of WORDNET (Miller, 1995) and FB15k is a subset of FREEBASE (Bollacker et al., 2008). We use the train, validation, and test split proposed by Bordes et al. (2013). The experiments on knowledge hypergraph completion are

conducted on three datasets. The first is JF17K proposed by Wen et al. (2016); as no validation set is proposed for JF17K, we randomly select 20% of the train set as validation. We also create two datasets FB-AUTO and M-FB15K from FREEBASE. See Appendix A for more dataset details.

## 5.2 BASELINES

To compare our results to that of existing work, we first design a few simple baselines that extend current models to work with knowledge hypergraphs. We only consider models that admit a simple extension to beyond-binary relations for the link prediction task. The baselines for this task are grouped into the following categories: (1) methods that work with binary relations and that are easily extendable to higher-arity: r-SimplE, m-DistMult, and m-CP; (2) existing methods that can handle higher-arity relations: m-TransH. Below we give some details about methods in category (1).

**r-SimplE:** To test performance of a model trained on reified data, we converted higher-arity relations in the train set to binary relations through reification. We then use the SimplE model (that we call r-SimplE) to train and test on this reified data. In this setting, at test time higher-arity relations are first reified to a set of binary relations; this process creates new auxiliary entities for which the model has no learned embeddings. To embed the auxiliary entities for the prediction step, we use the observation we have about them at test time. For example, a higher-arity relation $r(e_1, e_2, e_3)$ is reified at test time by being replaced by three facts: $r_1(id123, e_1)$, $r_2(id123, e_2)$, and $r_3(id123, e_3)$. When predicting the tail entity of $r_1(id123, ?)$, we use the other two reified facts to learn an embedding for entity $id123$. Because $id123$ is added only to help represent the higher-arity relations as a set of binary relations, we only do tail prediction for reified relations.

**m-DistMult:** DistMult (Yang et al., 2015) defines a score function $\phi(r(e_i, e_j)) = \odot(\mathbf{r}, \mathbf{e_i}, \mathbf{e_j})$. To accommodate non-binary relations, we redefine this function as $\phi(r(e_i, \ldots, e_j)) = \odot(\mathbf{r}, \mathbf{e_i}, \ldots, \mathbf{e_j})$.

**m-CP:** Canonical Polyadic (CP) decomposition (Hitchcock, 1927) embeds each entity $e$ as two vectors $\mathbf{e^{(1)}}$ and $\mathbf{e^{(2)}}$, and each relation $r$ as a single vector $\mathbf{r}$. CP defines the score function $\phi(r(e_i, e_j)) = \odot(\mathbf{r}, \mathbf{e_i^{(1)}}, \mathbf{e_j^{(2)}})$. We extend CP to a variant (m-CP) that accommodates non-binary relations, and which embeds each entity $e$ as $\delta$ different vectors $\mathbf{e^{(1)}}, .., \mathbf{e^{(\delta)}}$, where $\delta = \max_{r \in \mathcal{R}}(|r|)$. m-CP computes the score of a tuple as $\phi(r(e_i, \ldots, e_j)) = \odot(\mathbf{r}, \mathbf{e_i^{(1)}}, ..., \mathbf{e_j^{(|\mathbf{r}|)}})$.

## 5.3 EVALUATION METRICS

Given a knowledge hypergraph on $\tau'$, we evaluate various completion methods using a train and test set $\tau'_{train}$ and $\tau'_{test}$. We use two evaluation metrics: Hit@t and Mean Reciprocal Rank (MRR). Both these measures rely on the *ranking* of a tuple $x \in \tau'_{test}$ within a set of *corrupted* tuples. For each tuple $r(e_1, \ldots, e_k)$ in $\tau'_{test}$ and each entity position $i$ in the tuple, we generate $|\mathcal{E}| - 1$ corrupted tuples by replacing the entity $e_i$ with each of the entities in $\mathcal{E} \setminus \{e_i\}$. For example, by corrupting entity $e_i$, we would obtain a new tuple $r(e_1, \ldots, e_i^c, \ldots, e_k)$ where $e_i^c \in \mathcal{E} \setminus \{e_i\}$. Let the set of corrupted tuples, plus $r(e_1, \ldots, e_k)$, be denoted by $\theta_i(r(e_1, \ldots, e_k))$. Let $\text{rank}_i(r(e_1, \ldots, e_k))$ be the ranking of $r(e_1, \ldots, e_k)$ within $\theta_i(r(e_1, \ldots, e_k))$ based on the score $\phi(x)$ for each $x \in \theta_i(r(e_1, \ldots, e_k))$. In an ideal knowledge hypergraph completion method, the rank $\text{rank}_i(r(e_1, \ldots, e_k))$ is 1 among all corrupted tuples. We compute the MRR as $\frac{1}{K} \sum_{r(e_1, \ldots, e_k) \in \tau'_{test}} \sum_{i=1}^{k} \frac{1}{\text{rank}_i r(e_1, \ldots, e_k)}$ where $K = \sum_{r(e_1, \ldots e_k) \in \tau'_{test}} |r|$ is the number of prediction tasks. Hit@t measures the proportion of tuples in $\tau'_{test}$ that rank among top $t$ in their corresponding corrupted sets. We follow Bordes et al. (2013) and remove all corrupted tuples that are in $\tau'$ from our computation of MRR and Hit@t.

## 6 EXPERIMENTS

This section summarizes our experiments with HSimplE and HypE. We evaluate both models on knowledge hypergraphs, as well as on knowledge graphs, and show results on training with different embedding dimensions. Moreover, to test their representation power further, we evaluate HSimplE and HypE on a more challenging dataset that we describe below. We also conduct ablation studies based on performance breakdown across different arities.

Table 1: Knowledge hypergraph completion results on JF17K, FB-AUTO and M-FB15K for baselines and the proposed method. The prefixes 'r' and 'm' in the model names stand for *reification* and *multi-arity* respectively. Both our methods outperform the baselines on all datasets.

| | JF17K | | | | FB-AUTO | | | | M-FB15K | | | |
|---|---|---|---|---|---|---|---|---|---|---|---|---|
| Model | MRR | Hit@1 | Hit@3 | Hit@10 | MRR | Hit@1 | Hit@3 | Hit@10 | MRR | Hit@1 | Hit@3 | Hit@10 |
| r-SimplE | 0.102 | 0.069 | 0.112 | 0.168 | 0.106 | 0.082 | 0.115 | 0.147 | 0.051 | 0.042 | 0.054 | 0.070 |
| m-DistMult | 0.460 | 0.367 | 0.510 | 0.635 | 0.784 | 0.745 | 0.815 | 0.845 | 0.705 | 0.633 | 0.740 | 0.844 |
| m-CP | 0.392 | 0.303 | 0.441 | 0.560 | 0.752 | 0.704 | 0.785 | 0.837 | 0.680 | 0.605 | 0.715 | 0.828 |
| m-TransH (Wen et al., 2016) | 0.446 | 0.357 | 0.495 | 0.614 | 0.728 | 0.727 | 0.728 | 0.728 | 0.623 | 0.531 | 0.669 | 0.809 |
| HSimplE (Ours) | 0.472 | 0.375 | 0.523 | 0.649 | 0.798 | 0.766 | 0.821 | 0.855 | 0.730 | 0.664 | 0.763 | 0.859 |
| HypE (Ours) | **0.492** | **0.409** | **0.533** | **0.650** | **0.804** | **0.774** | **0.823** | **0.856** | **0.777** | **0.725** | **0.800** | **0.881** |

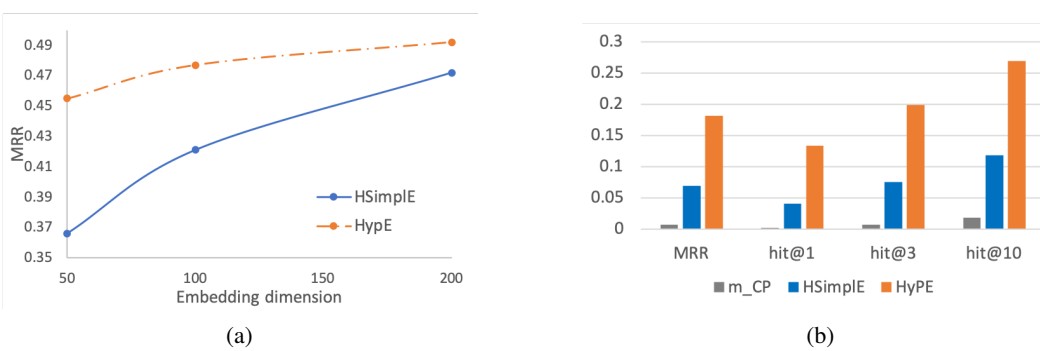

(a)                                          (b)

Figure 3: The above experiments show that HypE outperforms HSimplE when trained with fewer parameters, and when tested on samples that contain at least one entity in a position never encountered during training. (a) MRR of HypE and HSimplE for different embedding dimensions. (b) Results of m-CP, HSimplE, and HypE on the *missing positions* test set.

## 6.1 KNOWLEDGE HYPERGRAPH COMPLETION RESULTS

The results of our experiments, summarized in Table 1, show that both HSimplE and HypE outperform the proposed baselines across the three datasets JF17K, FB-AUTO, and M-FB15K. They further demonstrate that reification for the r-SimplE model does not work well; this is because the reification process introduces auxiliary entities for which the model does not learn appropriate embeddings because these auxiliary entities appear in very few facts. Comparing the results of r-SimplE against HSimplE, we can also see that extending a model to work with hypergraphs works better than reification when high-arity relations are present.

The ability of knowledge sharing through the learned position-dependent convolution filters suggests that HypE would need a lower number of parameters than HSimplE in order to obtain good results. To test this, we train both models with embedding dimension of 50, 100, and 200. Figure 3a shows the MRR evaluation on the test set for each model with different embedding sizes. Based on the MRR result, we can see that HypE outperforms HSimplE by 24% for embedding dimension 50, implying that HypE works better under a constrained budget. This difference becomes negligible for embedding dimensions of 200.

Disentangling the representations of entity embeddings and positional filters enables HypE to better learn the role of position within a relation, because the learning process considers the behaviour of all entities that appear in a given position at time of training. This becomes specially important in the case when some entities never appear in certain positions in the train set, but you still want to be able to reason about them no matter what position they appear in at test time. In order to test the effectiveness of our models in this more challenging scenario, we created a *missing positions* test set by selecting the tuples from our original test set that contain at least one entity in a position it never appears in in the train dataset. The results on these experiments (Figure 3b) show that (1) both HSimplE and HypE outperform m-CP (which learns different embeddings for each entity-position pair), and more importantly, (2) HypE significantly outperforms HSimplE for this challenging test set, leading us to believe that disentangling entity and position representations may be a better strategy for this scenario.

Table 2: Knowledge graph completion results on WN18 and FB15K for baselines and HypE. Note that we do not include results for HSimplE because for knowledge graphs, HSimplE is equivalent to SimplE. The table shows that HypE performs similar to the best baselines for knowledge graphs with binary relations.

| Model | WN18 | | | | FB15k | | | |
|---|---|---|---|---|---|---|---|---|
| | MRR | Hit@1 | Hit@3 | Hit@10 | MRR | Hit@1 | Hit@3 | Hit@10 |
| CP (Hitchcock, 1927) | 0.074 | 0.049 | 0.080 | 0.125 | 0.326 | 0.219 | 0.376 | 0.532 |
| TransH (Wang et al., 2014) | - | - | - | 0.867 | - | - | - | 0.585 |
| m-TransH (Wen et al., 2016) | 0.671 | 0.495 | 0.839 | 0.923 | 0.351 | 0.228 | 0.427 | 0.559 |
| DistMult (Yang et al., 2015) | 0.822 | 0.728 | 0.914 | 0.936 | 0.654 | 0.546 | 0.733 | 0.824 |
| SimplE (Kazemi & Poole, 2018) | **0.942** | **0.939** | **0.944** | **0.947** | **0.727** | **0.660** | 0.773 | 0.838 |
| HypE (Ours) | 0.934 | 0.927 | 0.940 | 0.944 | 0.725 | 0.648 | **0.777** | **0.856** |

Table 3: Breakdown performance of Hit@10 across relations with different arities on JF17K.

| Model | Arity | | | | | All |
|---|---|---|---|---|---|---|
| | 2 | 3 | 4 | 5 | 6 | |
| r-SimplE | **0.478** | 0.025 | 0.015 | 0.022 | 0.000 | 0.168 |
| m-DistMult | 0.359 | 0.591 | 0.745 | 0.869 | 0.359 | 0.635 |
| m-CP | 0.305 | 0.517 | 0.679 | 0.870 | 0.875 | 0.560 |
| m-TransH (Wen et al., 2016) | 0.316 | 0.563 | 0.762 | 0.925 | **0.979** | 0.614 |
| HSimplE (Ours) | 0.376 | 0.625 | 0.742 | 0.810 | 0.010 | 0.649 |
| 'HypE (Ours) | 0.338 | **0.626** | **0.776** | **0.936** | 0.948 | **0.650** |

## 6.2 KNOWLEDGE GRAPH COMPLETION RESULTS

To confirm that HSimplE and HypE still work well on the more common knowledge graphs, we evaluate them on WN18 and FB15K. Table 2 shows link prediction results on WN18 and FB15K. Baseline results are taken from the original papers except that of m-TransH, which we implement ourselves. Instead of tuning the parameters of HypE to get potentially better results, we instead follow the Kazemi & Poole (2018) setup with the same grid search approach by setting $n = 2$, $l = 2$, and $s = 2$. This results in all models in Table 2 having the same number of parameters, and thus makes them directly comparable to each other. Note that since HSimplE is equivalent to SimplE for binary relations (as shown in Section 4), we have excluded HSimplE from the table. The results show that on WN18 and FB15K, HypE outperforms all baselines except SimplE, while its performance remains comparable to that of SimplE.

## 6.3 ABLATION STUDY ON DIFFERENT ARITIES

We break down the performance of HSimplE, HypE and each of the baselines across relations with different arities. Table 3 shows the Hit@10 results of the models for each arity in JF17K. We observe that the proposed models outperform the state-of-the-art and the baselines in all arities except arity 6, which has a total of only 37 tuples in the train and test sets. Table 3 also shows that the performance of all models generally improve as arity increases. We note that the train set has much fewer relation types that are defined on a high number of entities — JF17K contains only two relation types that admit six entities. This leads us to hypothesize that the position and/or entity representations learned for higher arities are optimized for these few relation types.

## 7 CONCLUSIONS

Knowledge hypergraph completion is an important problem that has received little attention. In this work, having introduced two new knowledge hypergraph dataset, baselines, and two new methods for link prediction in knowledge hypergraphs, we hope to kindle interest in the problem. Unlike knowledge graphs, hypergraphs have a more complex structure that opens the door to more challenging questions such as: how do we effectively predict the missing entities in a given (partial) tuple? Is MRR a good evaluation metric for hypergraphs?

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

## A  APPENDIX

### A.1  DATASETS

Table 4 summarizes the statistics of the datasets. Note first that FREEBASE is a reified dataset; that is, it is created from a knowledge base having facts with relations defined on two or more entities. To obtain a knowledge hypergraph $H$ from FREEBASE, we perform an inverse reification process by following the steps below.

1. From FREEBASE, remove the facts that have relations defined on a single entity, or that contain numbers or enumeration as entities.

2. Convert the triples in FREEBASE that share the same entity into facts in $H$. For example, the triples $r_0(id123, e_i)$, $r_1(id123, e_j)$, and $r_2(id123, e_k)$, which were originally created by the addition of the (unique) reified entity $id123$, now represent fact $r(e_i, e_j, e_k)$ in $H$.

3. Create the FB-AUTO dataset by selecting the facts from $H$ whose subject is 'automotive'.

4. Create the M-FB15K dataset by following a strategy similar to that proposed by Bordes et al. (2013): select the facts in $H$ that pertain to entities present in the Wikilinks database (Singh et al., 2012).

5. Split the facts in each of FB-AUTO and M-FB15K randomly into train, test, and validation sets.

Table 4: Dataset Statistics.

| Dataset | $|\mathcal{E}|$ | $|\mathcal{R}|$ | #train | #valid | #test | #arity=2 | #arity=3 | #arity=4 | #arity=5 | #arity=6 |
|---------|------|------|--------|--------|-------|----------|----------|----------|----------|----------|
| WN18 | 40,943 | 18 | 141,442 | 5,000 | 5,000 | 151,442 | 0 | 0 | 0 | 0 |
| FB15k | 14,951 | 1,345 | 483,142 | 50,000 | 59,071 | 592,213 | 0 | 0 | 0 | 0 |
| JF17K | 29,177 | 327 | 77,733 | – | 24,915 | 56,322 | 34,550 | 9,509 | 2,230 | 37 |
| FB-AUTO | 3,410 | 8 | 6,778 | 2,255 | 2,180 | 3,786 | 0 | 215 | 7,212 | 0 |
| M-FB15K | 10,314 | 71 | 415,375 | 39,348 | 38,797 | 82,247 | 400,027 | 26 | 11,220 | 0 |

## A.2 IMPLEMENTATION DETAILS

We implement HSimplE, HypE and the baselines in PyTorch (Paszke et al., 2017). We use Adagrad (Duchi et al., 2011) as the optimizer and dropout (Srivastava et al., 2014) to regularize our model and baselines. We tune our hyperparameters over the validation set, and fix the maximum number of epochs to 500 and batch size to 128. We set the embedding size and negative ratio to 200 and 10 respectively. We compute the MRR of models over the validation set every 50 epochs and select the epoch that results the best. The learning rate and dropout rate of all models are tuned. HypE also has $n$, $l$ and $s$ as hyperparameters. We select the hyperparameters of HypE, HSimplE and baselines via the same grid search based on MRR on the validation. The code of the proposed model, the baselines, and the datasets are available in this link.

## A.3 FULL EXPRESSIVITY

**Theorem 1 (Expressivity of HypE)** *Let $\tau$ be a set of true tuples defined over entities $\mathcal{E}$ and relations $\mathcal{R}$, and let $\delta = \max_{r \in \mathcal{R}}(|r|)$ be the maximum arity of the relations in $\mathcal{R}$. There exists a HypE model with embedding vectors of size at most $\max(\delta|\tau|, \delta)$ that assigns 1 to the tuples in $\tau$ and 0 to others.*

**Proof:** To prove the theorem, we show an assignment of embedding values for each of the entities and relations in $\tau$ such that the scoring function of HypE is as follows:

$$\phi(x) = \begin{cases} 0 & \text{if } x \in \tau \\ 1 & \text{otherwise} \end{cases}$$

We begin the proof by first describing the embeddings of each of the entities and relations in HypE; we then proceed to show that with such an embedding, HypE can represent any world accurately.

Let us first assume that $|\tau| > 0$ and let $f_p$ be the $p$th fact in $\tau$. We let each entity $e \in \mathcal{E}$ be represented with a vector of length $\delta|\tau|$ in which the $p$th block of $\delta$-bits is the one-hot representation of $e$ in fact $f_p$: if $e$ appears in fact $f_p$ at position $i$, then the $i$th bit of the $p$th block is set to 1, and to 0 otherwise. Each relation $r \in \mathcal{R}$ is then represented as a vector of length $\tau$ whose $p$th bit is equal to 1 if fact $f_p$ is defined on relation $r$, and 0 otherwise.

HypE defines different convolutional weight filters for each entity position within a tuple. As we have at most $\delta$ possible positions, we define each convolutional filter $\omega_i$ as a vector of length $\delta$ where the $i$th bit is set to 1 and all others to 0, for each $i = 1, 2, \ldots, \delta$. When the scoring function $\phi$ is applied to some tuple $x$, for each entity position $i$ in $x$, convolution filter $\omega_i$ is applied to the entity at position $i$ in the tuple as a first step; the $\odot()$ function is then applied to the resulting vector and the relation embedding to obtain a score.

Given any tuple $x$, we want to show that $\phi(x) = 1$ if $x \in \tau$ and 0 otherwise.

First assume that $x = f_p$ is the $p$th fact in $\tau$ that is defined on relation $r$ and entities where $e_i$ is the entity at position $i$. Convolving each $e_i$ with $\omega_i$ results in a vector of length $|\tau|$ where the $p$th bit is equal to 1 (since both $\omega_i$ and the $p$th block of $e_i$ have a 1 at the $i$th position) (See Figure 4. Then, as a first step, function $\odot()$ computes the element-wise multiplication between the embedding of relation $r$ (that has 1 at position $p$) and all of the convolved entity vectors (each having 1 at position

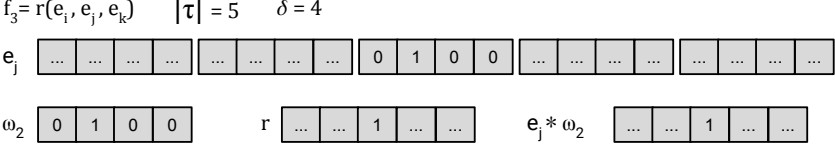

Figure 4: An example of an embedding where $|\tau| = 5$, $\delta = 4$ and $f_3$ is the third fact in $\tau$

$p$); this results in a vector of length $|\tau|$ where the $p$th bit is set to 1 and all other bits set to 0. Finally, $\odot(())$ sums the outcome of the resulting products to give us a score of 1.

To show that $\phi(x) = 0$ when $x \notin \tau$, we prove the contrapositive, namely that if $\phi(x) = 1$, then $x$ must be a fact in $\tau$. We proceed by contradiction. Assume that there exists a tuple $x \notin \tau$ such that $\phi(x) = 1$. This means that at the time of computing the element-wise product in the $\odot()$ function, there was a position $j$ at which all input vectors to $\odot()$ had a value of 1. This can happen only when (1) applying the convolution filter $w_j$ to each of the entities in $x$ produces a vector having 1 at position $j$, and (2) the embedding of relation $r \in x$ has 1 at position $j$.

The first case can happen only if all entities of $x$ appear in the $j$th fact $f_j \in \tau$; the second case happens only if relation $r \in x$ appears in $f_j$. But if all entities of $x$ as well as its relation appear in fact $f_j$, then $x \in \tau$, contradicting our assumption. Therefore, if $\phi(x) = 1$, then $x$ must be a fact in $\tau$.

To complete the proof, we consider the case when $|\tau| = 0$. In this case, since there are no facts, all entities and relations are represented by zero-vectors of length $\delta$. Then, for any tuple $x$, $\phi(x) = 0$. This completes the proof. $\qquad\square$

**Theorem 2 (Expressivity of HSimplE)** *Let $\tau$ be a set of true tuples defined over entities $\mathcal{E}$ and relations $\mathcal{R}$, and let $\delta = \max_{r \in \mathcal{R}}(|r|)$ be the maximum arity of the relations in $\mathcal{R}$. There exists a HSimplE model with embedding vectors of size at most $\max(\delta|\tau|, \delta)$ that assigns 1 to the tuples in $\tau$ and 0 to others.*

The proof of Theorem 2 is similar to the proof of Theorem 1, except that instead of applying convolution filters, shifting is applied on entity embeddings. After applying shifting, the same proof by contradiction holds for HSimplE.

