# OpenReview forum: "Knowledge Hypergraphs: Prediction Beyond Binary Relations"
_ICLR.cc/2020/Conference — Reject_

### Official Review · AnonReviewer1 · 2019-10-24
**Official Blind Review #1**

**Rating:** 3

**Review:**

Contributions:
1. This paper extends SimplE, a previous embedding model, from modeling binary knowledge graphs to knowledge hypergraphs, where n-ary relations may show up.
2. The paper designs two different architectures, HypE and HSimplE, to achieve knowledge hypergraph embedding.
3. Empirical results comparing with existing methods are proposed.

The major contribution of this paper is to propose two new architectures for hypergraph embedding. However, I still have some concerns after reading this paper.

1. The paper proposes to use 1d convolutions to separate entity positions and entity embeddings (HypE) and claim this strategy is better than just rotating the entity embeddings (HsimplE) because one can introduce additional parameters through the convolution filters. I'm expecting to see more evidence justifying this strategy. For example, in Table 2, SimplE is better than HypE in most cases. So probably HypE is not the most efficient way to use parameters. Besides, I'm curious about how performance will be when you use different n,l,d,s, since some of them are set to be very small according to 6.2.

2. The second thing that I feel confused about is the fair comparison with other methods to handle hypergraphs. For example, t-SimplE performs badly on all metrics, including Hit@t and MRR in Table 1. This could result from a sub-optimal adaptation/reification of SimplE to hypergraphs. However, in Table 3 it does much better than the rest methods for binary relations (0.478 vs others). This makes me confused and I don't understand why that could happen. Moreover, have you tried another adaptation method? For example, by converting non-binary relations into cliques?

I think the paper makes some contribution to the existing literature, but it should at least clarify its contribution with more evidence and better comparison criterions.

**Experience Assessment:**

I have read many papers in this area.

**Review Assessment: Checking Correctness Of Derivations And Theory:**

I carefully checked the derivations and theory.

**Review Assessment: Checking Correctness Of Experiments:**

I carefully checked the experiments.

**Review Assessment: Thoroughness In Paper Reading:**

I read the paper at least twice and used my best judgement in assessing the paper.

---

> ### Author Response · Authors · 2019-11-08
> **Response to Reviewer 1 - Part 1**
>
> Thanks for your constructive feedback. Here are responses to your question/concerns:
>
> -“The paper proposes to use 1d convolutions to separate entity positions and entity embeddings (HypE) and claim this strategy is better than just rotating the entity embeddings (HsimplE) because one can introduce additional parameters through the convolution filters. I'm expecting to see more evidence justifying this strategy."
>
> In general, HypE (that separates entity/position embeddings) works better than HSimple (that rotates entity embeddings) and this is evidenced in the paper by the following.
> When the knowledge base contains samples having variable size arity, then HypE performs consistently better (evidence in Table 1).
> When we vary the number of parameters of the model, HypE performs consistently better than HSimplE (evidence in Figure 3(a)).
> HSimple performs significantly worse (although not as bad as m-CP) when tested on a tuple with an entity appearing in a position that the model never encountered during train time. For example, testing the tuple r(e_1, e_2) with HSimplE will produce poor results if at train time HSimplE never encountered a sample where e_1 appears in position 1 (or equivalently, e_2 appears in position 2). This is evidenced by Figure 3(b).
>
> Given all the above experimental analyses, we conclude that in general, HypE is a better model than HSimplE, and disentangling entity representations from the position they appear in a relation is a better strategy.
>
> -"For example, in Table 2, SimplE is better than HypE in most cases. So probably HypE is not the most efficient way to use parameters. Besides, I'm curious about how performance will be when you use different n,l,d,s, since some of them are set to be very small according to 6.2.”
>
> For the specific case of airty 2, we first highlight that the average performance gain of SimplE over HypE across all metrics/datasets in Table 2 is a mere 0.2%
>
> An explanation of this slight difference may lie in the fact that, given that our primary interest in this paper is in solving the more general task of hypergraph completion, we did not spend too much energy fine tuning the hyperparameters of HypE (note that the reported results of SimplE are the ones reported in their paper, which was tuned for that case). Instead, we concentrated on making sure that the parameters in HypE are comparable to that of SimplE, namely that  n = 2, l = 2, and s = 2.
>
> Having said this, however, we do welcome the question asked by the reviewer and we will run experiments with a range of parameter values; we will report the results once we have them.

---

> ### Author Response · Authors · 2019-11-08
> **Response to Reviewer 1 - Part 2**
>
> -"Why does r-SimplE perform badly on arity > 2 but well on arity = 2?"
>
> r-SimplE consists of applying SimplE on a reified knowledge hypergraph. Reification converts the knowledge hypergraph into a knowledge graph where all relations are binary. The main issue with reification is that converting relations with arity > 2 into binary ones introduces new entities not seen during training (square nodes in Figure 3(b)), while arity 2 relations remain unchanged in the process. Accordingly, r-SimplE on arity 2 relations behaves exactly as SimplE, with good test results. On the other hand, relations that had higher arity in the original graph now contain new entities that the model needs to handle:
> To embed the new entities at test time, we use the observation we have about them. For example, a higher-arity relation r(e1, e2, e3) is reified to three facts: r1(id123, e1), r2(id123, e2), and r3(id123, e3). When predicting the tail entity of r1 (id123, ?), we use the other two reified facts to learn an embedding for entity id123 (See Section 5.2 for more details). This is the key reason why the performance of r-SimplE drops: r-SimplE performs badly on all the metrics of Table 1 because it performs poorly on arities higher than 2. Table 3 explains how r-SimplE performs for each arity showing the expected behavior: works well for arity 2 and poorly for higher arity relations.
>
> -"Could the results in Table 1 be the outcome of “a sub-optimal adaptation/reification of SimplE to hypergraphs”?"
>
> 1) As described above, we conclude that the main reason why r-SimplE performs poorly on relations with arity > 2 is because the model struggles to find an appropriate embedding of the newly created entities (square nodes in Figure 3(b)). As is the case in knowledge graphs, entities do not have attributes and the only information we have about the square nodes is who their neighbors are. This is the information we use in r-SimplE to embed the new nodes, and obtain the (poor) results in Table 3. We do not see much room for optimizing this step in a way that would produce any significant improvement to the model.
>
> 2) Reification is a deterministic process (different reifications of the same hypergraph produces the same graph up to renaming every time), and thus there is no question of optimization (See Section 2).
>
> -"Moreover, have you tried another adaptation method? For example, by converting non-binary relations into cliques?”
>
> We considered the star-to-clique method of converting hypergraphs into graphs early on in the project, but discarded it because of the following two reasons (that are discussed in Section 2):
>
> 1) Star-to-clique cannot be tested sensibly as it only makes predictions about the pairs, not about the tuples. So it makes many more false positive predictions, but the mechanism we use for testing - without negative test cases - does not adequately evaluate such predictions. Thus, while the method we use for testing is reasonable for other methods, it is not a fair evaluation of star-to-clique.
>
> 2) Unlike in the case of reification, converting non-binary relations into cliques causes a loss of information (as described in Section 2 and Figure 3). This would mean that, to begin with, the model would not have all the information needed to learn to make predictions. Thus, we do not expect a model trained on partial information to perform better (or as well as) a model trained on the full dataset.

---

### Official Review · AnonReviewer3 · 2019-10-26
**Official Blind Review #3**

**Rating:** 6

**Review:**

This paper proposes two new embedding strategies for the task of knowledge graph completion, with special attention on generalizations that support hypergraphs.  The first method, HSimplE, learns an embedding for entities that directly contains multiple positional representations; these are shifted depending on the relation they're used in.  The second method, HypE, disentangles entity embeddings and positional convolutional filters, allowing stronger positional generalization.  Experiments on standard benchmarks demonstrate that the approaches work well.

I think this paper should be accepted.  While the ideas are so simple that they border on being trivial generalizations of previous work, the paper is well written, and the results seem solid.  I think this is work that needs to be done, so I favor accepting it.

On the positive side:

* The ideas underlying HSimplE and HypE are natural and clear - we basically want to learn better embeddings for hypergraphs, and there are a couple of obvious ways to do that.  Both of these seem clear.

* The experiments are nicely done, and show a consistent (if unsurprising) benefit to the approach.

* The paper is well written and well situated in the literature.

* I expect that other researchers in this area will be able to reproduce and build upon this work without any difficulty.

On the negative side:

* The ideas are obvious; the results are unsurprising; the paper lacks "deep insight".  It is a contribution in the sense that someone needed to do this work, and I'm glad that it's been done (and done well), but it's not earth-shattering.

* It doesn't seem like this is quite the best version of this idea.  I really like the idea of HypE, but it seems strange that entity embeddings are modulated based on position *only*, without regard to relation -- that is, it seems like a given entity in position #2 might need to be represented in very different ways depending on which relation is being used.

**Experience Assessment:**

I have read many papers in this area.

**Review Assessment: Checking Correctness Of Derivations And Theory:**

I carefully checked the derivations and theory.

**Review Assessment: Checking Correctness Of Experiments:**

I carefully checked the experiments.

**Review Assessment: Thoroughness In Paper Reading:**

I read the paper thoroughly.

---

> ### Author Response · Authors · 2019-11-08
> **Thanks for your feedback**
>
> Thanks for your constructive feedback. Here are responses to your questions/concerns.
>
> -”I expect that other researchers in this area will be able to reproduce and build upon this work without any difficulty.”
>
> The code for all baselines and proposed models and all the datasets are available in an anonymized link in OpenReview. Two other groups of researchers also already reproduced these results.
>
> -”It doesn't seem like this is quite the best version of this idea.  I really like the idea of HypE, but it seems strange that entity embeddings are modulated based on position *only*, without regard to relation -- that is, it seems like a given entity in position #2 might need to be represented in very different ways depending on which relation is being used.”
>
> This was a very interesting observation that we too wondered about early on in the project. In fact, one of the first set of experiments we tried was to learn the entity modulating filters based on *both* position and relation type. In all our experiments, the results showed that this variant always produced slightly worse results than using simply position to modulate the entity embeddings.
> Based on these experiments (that we decided to exclude due to limited space) our conclusion was that making filters both relation and position-dependent introduced even more parameters, and given that our knowledge graphs are sparse in nature, this meant we did not have enough samples to efficiently learn them all.

---

### Official Review · AnonReviewer2 · 2019-10-28
**Official Blind Review #2**

**Rating:** 1

**Review:**

The paper proposes models for link prediction task in a generalized knowledge graph setting that can contain n-ary
relationships. The paper explains the problem of using only binary relationships to model real-world scenarios, typical workaround for converting n-ary relationships to binary relationships and their
problems. The work then extends one model for link prediction in a binary setting to n-ary setting. Work also proposes a novel model using positional convolutional filters for entity embeddings to model n-ary relationships and use that for link prediction. Work also establishes baselines for the new problem and publishes two
datasets (subsets of FREEBASE) for the same. Experimental results show the proposed models, outperform the baselines by good margin (with some exceptions).

This is a very obvious generalization from binary to n-array relations, but the work is very incremental and does not provide any good motivations. It is well known that hypergraphs can be approximated with graphs using clique and star expansions but I do not see any discussion regarding this. Frankly, I do not see any good motivation to consider this generalization.

 Comments:
 1. Table 3: for arity 2, r-simply is performing better, but HSimplIE is marked best. Ideally, for arity 2 both should
have the same result right?
 2. For arity 6, there is a sudden drop in performance for both the proposed models. But the baseline model mtransH holds very tight. Any reason or explanation for this massive drop and large gap in performance.
 3. Have you conducted experiments for higher arity? On the continuation of the above point, the performance drop consistently for higher arity than 5? '


**Experience Assessment:**

I have published one or two papers in this area.

**Review Assessment: Checking Correctness Of Derivations And Theory:**

I carefully checked the derivations and theory.

**Review Assessment: Checking Correctness Of Experiments:**

I assessed the sensibility of the experiments.

**Review Assessment: Thoroughness In Paper Reading:**

I read the paper thoroughly.

---

> ### Author Response · Authors · 2019-11-08
> **Thanks for your feedback**
>
> Thanks for your constructive feedback. Here are responses to your questions/concerns:
>
> -”It is well known that hypergraphs can be approximated with graphs using clique and star expansions but I do not see any discussions regarding this. Frankly, I do not see any good motivation to consider this generalization.”
>
> The paper does discuss adequately the problems of the mentioned hypergraph approximation techniques for knowledge graph completion; as the reviewer himself/herself states at the beginning of the review: “The paper explains the problem of using only binary relationships to model real-world scenarios, typical workaround for converting n-ary relationships to binary relationships and their problems.” Below we summarize the discussions explained in Section 2 and the baseline experiment in Section 5.2 and Section 6.1.
> Hypergraphs can be converted into graphs using star-to-clique or reification as extensively discussed in Section 2. Star-to-clique is an approximation that loses information which makes it difficult to use in practice. Reification does not lose information but introduces new entities that are difficult to handle. We show experimental results on how poorly reification works in Table 1.
>
> We discuss hypergraph conversions extensively in Section 2 and provide examples in Figure 1. As we explain in the section, both conversion methods have their disadvantages when it comes to using these graph structures to learn relational information. In addition, we show experimental results on how poorly the model performs when we train with a reified graph. See the r-SimplE baseline results in Table 1.
>
> We believe the paper adequately addresses the answer to this question.
>
> -”Table 3: for arity 2, r-simply is performing better, but HSimpIE is marked best. Ideally, for arity 2 both should have the same result right?”
>
> The marking is an error on our side, which we will fix in the final version. Thank you for pointing this out.
> It is not expected that HSimplE and r-SimplE have the same result for arity 2 as they are different models even for arity 2. r-SimplE consists of first converting the knowledge hypergraph into a knowledge graph using reification and then using the resulting graph to train SimplE (refer to Section 5.2 for more details). However, for arity-2, there is no reification happening since all edges already have arity 2. This implies that for this special case of arity 2, r-SimplE is the same model as SimplE.
> On the other hand, HSimplE is a more general model than SimplE, as it is designed to work with relations with arbitrary size arity. Since in our experiments both SimplE and HSimplE have the same number of parameters, but the latter is expected to learn a more general structure (arity > 2), it is not unusual that SimplE — tuned to this specific case — works better for arity 2.
>
> -”For arity 6, there is a sudden drop in performance for both the proposed models. But the baseline model mtransH holds very tight. Any reason or explanation for this massive drop and large gap in performance. Have you conducted experiments for higher arity? On the continuation of the above point, the performance drop consistently for higher arity than 5? '”
>
> The results reported in Table 4 for arity 6 are not reliable for any of the models, as there are only 37 samples total in the dataset (18 train, 2 validation, and 17 in test), as opposed to the more than a few thousand samples in each one of the remaining arities (refer to Table 4 for other details).  Even though we mention this fact in the paper, in hindsight we realize that we probably should include error bars or should have removed the arity 6 column from Table 4 in order not to mislead the reader; Accordingly, there is no evidence of a drop in performance for higher arities.

---

> > ### Author Response · Authors · 2019-11-13
> > **Update**
> >
> > Thanks to the reviewer’s observation, we looked again into the results for arity 6 that we reported previously in our paper and found that our reporting for HypE for airty 6 contains an editing error and that the results are better than we claimed: The correct Hit@10 score for HypE (arity 6) is 0.948. However, with only 18 training and 16 test tuples, the results for arity-6 cannot be treated as reliable.

---

### Decision · Program_Chairs · 2019-12-19

**Decision:**

Reject

**Comment:**

The paper proposes two methods for link prediction in knowledge hypergraphs. The first method concatenates the embedding of all entities and relations in a hyperedge. The second method combines an entity embedding, a relation embedding, and a weighted convolution of positions. The authors demonstrate on two datasets (derived by the authors from Freebase), that the proposed methods work well compared to baselines. The paper proposes direct generalizations of knowledge graph approaches, and unfortunately does not yet provide a comprehensive coverage of the possible design space of the two proposed extensions.

The authors should be commended for providing the source code for reproducibility. One of the reviewers (who was unfortunately also the most negative), was time pressed. Unfortunately, the discussion period was not used by the reviewers to respond to the authors' rebuttal of their concerns.

Even discounting the most negative review, this paper is on the borderline, and given the large number of submissions to ICLR, it unfortunately falls below the acceptance threshold in its current form.